# Reporting Guidelines for Whole-Body Vibration Studies in Humans, Animals and Cell Cultures: A Consensus Statement from an International Group of Experts

**DOI:** 10.3390/biology10100965

**Published:** 2021-09-27

**Authors:** Marieke J. G. van Heuvelen, Jörn Rittweger, Stefan Judex, Borja Sañudo, Adérito Seixas, Anselm B. M. Fuermaier, Oliver Tucha, Csaba Nyakas, Pedro J. Marín, Redha Taiar, Christina Stark, Eckhard Schoenau, Danúbia C. Sá-Caputo, Mario Bernardo-Filho, Eddy A. van der Zee

**Affiliations:** 1Department of Human Movement Sciences, University Medical Center Groningen, University of Groningen, 9713 AV Groningen, The Netherlands; 2Institute of Aerospace Medicine, German Aerospace Center (DLR), 51147 Cologne, Germany; joern.rittweger@dlr.de; 3Department of Pediatrics and Adolescent Medicine, University of Cologne, D50931 Cologne, Germany; eckhard.schoenau@uk-koeln.de; 4Department of Biomedical Engineering, Stony Brook University, Stony Brook, NY 11794, USA; stefan.judex@stonybrook.edu; 5Departamento de Educación Física y Deporte, Universidad de Sevilla, 41013 Seville, Spain; bsancor@us.es; 6Escola Superior de Saúde Fernando Pessoa, 4200-253 Porto, Portugal; aderito@ufp.edu.pt; 7Department of Clinical and Developmental Neuropsychology, University of Groningen, 9712 TS Groningen, The Netherlands; a.b.m.fuermaier@rug.nl (A.B.M.F.); o.m.tucha@rug.nl (O.T.); 8Department of Psychiatry and Psychotherapy, University Medical Center Rostock, 18147 Rostock, Germany; 9Department of Morphology and Physiology, Faculty of Health Sciences, Semmelweis University, H-1088 Budapest, Hungary; nyakas.csaba@tf.hu; 10Development Research, CYMO Research Institute, 47140 Valladolid, Spain; pjmarin@cymori.com; 11MATIM, Department of Sport Science, Université de Reims Champagne Ardenne, 51100 Reims, France; redha.taiar@univ-reims.fr; 12Department of Neurology, University of Cologne, D50931 Cologne, Germany; christina.stark@uk-koeln.de; 13Faculdade Bezerra de Araújo, Rio de Janeiro 23052-180, Brazil; dradanubia@gmail.com; 14Laboratório de Vibrações Mecânicas e Práticas Integrativas, Instituto de Biologia Roberto Alcântara Gomes and Policlínica Piquet Carneiro, Universidade do Estado do Rio de Janeiro, Rio de Janeiro 20950-003, Brazil; bernardofilhom@gmail.com; 15Molecular Neurobiology, Groningen Institute for Evolutionary Life Sciences (GELIFES), University of Groningen, 9747 AG Groningen, The Netherlands; e.a.van.der.zee@rug.nl

**Keywords:** checklist, EQUATOR, vibration exercise, vibration therapy, writing

## Abstract

**Simple Summary:**

Whole-body vibration (WBV) is an exercise or treatment method used in sports, physiotherapy, and rehabilitation. During WBV, people sit, stand, or exercise on a platform that generates vibrations. These vibrations generally occur between 20 and 60 times per second and have a magnitude of one or several millimeters. Research is focused on the effects of WBV on, for instance, physical and cognitive functions as well as the underlying mechanisms that may explain the effects. Research is not only done in humans but in animals and cell cultures as well. It is important to report the studies correctly, completely, and consistently. This way, researchers can interpret and compare each other’s studies, and data of different studies can be combined and analyzed together. To serve this goal, we developed new guidelines on how to report on WBV studies. The guidelines include checklists for human and animal/cell culture research, explanations, and examples of how to report. We included information about devices, vibrations, administration, general protocol, and subjects. The guidelines are WBV-specific and can be used by researchers alongside general guidelines for specific research designs.

**Abstract:**

Whole-body vibration (WBV) is an exercise modality or treatment/prophylaxis method in which subjects (humans, animals, or cells) are exposed to mechanical vibrations through a vibrating platform or device. The vibrations are defined by their direction, frequency, magnitude, duration, and the number of daily bouts. Subjects can be exposed while performing exercises, hold postures, sitting, or lying down. Worldwide, WBV has attracted significant attention, and the number of studies is rising. To interpret, compare, and aggregate studies, the correct, complete, and consistent reporting of WBV-specific data (WBV parameters) is critical. Specific reporting guidelines aid in accomplishing this goal. There was a need to expand existing guidelines because of continuous developments in the field of WBV research, including but not limited to new outcome measures regarding brain function and cognition, modified designs of WBV platforms and attachments (e.g., mounting a chair on a platform), and comparisons of animal and cell culture studies with human studies. Based on Delphi studies among experts and using EQUATOR recommendations, we have developed extended reporting guidelines with checklists for human and animal/cell culture research, including information on devices, vibrations, administration, general protocol, and subjects. In addition, we provide explanations and examples of how to report. These new reporting guidelines are specific to WBV variables and do not target research designs in general. Researchers are encouraged to use the new WBV guidelines in addition to general design-specific guidelines.

## 1. Introduction

Whole-body vibration (WBV) is an exercise modality or treatment/prophylaxis method in which subjects are exposed to mechanical vibrations through a vibrating platform. WBV has attracted significant attention in sports and rehabilitation, especially to increase muscle activity [1,2,3], improve muscle function [1,4], and enhance bone mass and morphology [5,6,7] and several other medical outcomes [8,9,10]. In addition, WBV appears beneficial for cognitive function [11,12,13,14]. However, despite the convincing evidence in favor of WBV, controversial findings have been found for specific outcomes [15,16], and its clinical relevance has been questioned for specific populations [17,18]. The two principal types of vibration transmissions are vertical vibration transmission and side-alternating or tilting vibration transmission (Figure 1). During WBV, subjects can perform dynamic exercises on the platform (e.g., doing back squats) or take a static posture like standing upright, sitting, or lying down. Furthermore, the intensity of WBV is defined by the frequency, magnitude (i.e., the wave ‘size’), wave form (sinusoidal or not), exposure time, and the number of daily bouts [19].

The beneficial effects of WBV have been predominantly examined in clinical studies on humans within the fields of, e.g., sport science, human movement science, medicine, and physical therapy. Additionally, preclinical vibration studies with translational potential to clinical situations have been performed. These in vivo or in vitro preclinical studies are critical towards identifying putative mechanisms by which tissues and cells perceive and respond to the biomechanical signal. They will also aid in forming hypotheses regarding promising outcome measures and effective vibration settings. For in vitro cell culture studies, inherently, the term WBV is not appropriate. Nevertheless, in the current paper, we will use the term WBV for animal and cell culture studies as well if the study aim is to contribute knowledge towards WBV as an exercise modality or prophylaxis/treatment and if the vibration is not applied locally to a single muscle or bone.

With the increasing popularity of WBV, the yearly number of publications has quadrupled in the last 15 years. However, this growing number of publications may challenge the quality of reporting. Correct, complete, and consistent reporting is critical for several reasons. First, sufficient detail about what was planned and done in the study is needed for the correct interpretation of study results. Strengths and weaknesses in study design, execution of the experiment, and analyses should be given in order to judge the reliability of the results. To compare studies, common language and consensus about terminology are also needed. This will facilitate the understanding of inconsistent results and the judgment of whether study results can be aggregated in systematic reviews and meta-analyses. Correct, complete, and consistent reporting is also needed to adequately replicate studies and build on knowledge for further scientific development in the field of WBV. In addition, ethical reasons play a role here [20]. The integrity of the individual researcher requires honesty and completeness. Beyond this level, society is entitled to adequate information to minimize the risk of harm and to maximize the benefits for the final users of the data. Finally, the fair use of resources, funding, and time should be warranted in order not to waste money [21].

WBV studies are sensitive to inadequate reporting because of their technical approach, with different brands of platforms, types of vibration transmission and settings, the variety of applications, and the interdisciplinarity of the research. Lorenzen et al. [22] identified inconsistent use of terminology, especially regarding the magnitude of the vibration expressed as peak-to-peak displacement, peak acceleration, and/or root-mean-square (rms) acceleration. With respect to its application, in a recent review, 42% of studies failed to report the use of footwear during WBV [23] despite the fact that this item was included in the recommendations of Rauch et al. [24]. The current study is done to increase the visibility of reporting guidelines for WBV studies, to promote its use, and to extend the previously published recommendations [24]. The latter is justified by new developments in the field of WBV research, including, but not limited to, new outcome measures regarding brain function and cognition, modified designs of WBV platforms and attachments (e.g., mounting a chair on a platform), and comparisons of animal and cell culture studies with human studies. In addition, we use the recommendations of the EQUATOR (Enhancing the QUAlity and Transparency Of health Research) Network [25] to update and further develop reporting guidelines for WBV studies in order to enhance the quality and impact of the guidelines. Since only 6% of WBV publications from 2011 to February 2021, included in PubMed and Web of Science, cited the paper of Rauch et al. [24], it is important to stimulate a widespread use of the guidelines as well.

### Aim

The aim of this study is to expand the reporting guidelines for WBV studies in accordance with EQUATOR methodology for the development of guidelines. The guidelines focus on WBV-specific aspects and supplement general guidelines such as the CONSORT statement for reporting randomized controlled trials [26], the STROBE statement for reporting observational studies [27], and the ARRIVE guideline for in vivo animal research [28]. The guidelines aim to be used by researchers, peer reviewers, and editors.

## 2. Methods

We used the recommendations of the EQUATOR Network. The EQUATOR Network is an international initiative seeking to improve the quality of scientific publications by promoting transparent and accurate reporting [25]. The EQUATOR Network [29] provides a library with a collection of guidelines used in health research and a set of toolkits with, among others, a strategy for how to develop guidelines. We followed the steps described in this toolkit to update and develop our reporting guidelines. We used the infrastructure of the international World Association of Vibration Experts (WAVEX) for meetings and collaboration. WAVEX was set up in 2016 in Rio de Janeiro (Brazil) and is currently hosted by the University of Groningen (The Netherlands). The aims of WAVEX are to share knowledge, promote collaboration and enhance the quality of WBV research. Members come from all over the world and meet regularly during an international conference.

Thus, the steps we followed were:1.First face-to-face expert meeting. During the WAVEX Conference in Groningen (23–24 August 2018), we had face-to-face meetings with nine experts. We discussed the need for complete, visible, and widely used reporting guidelines specific to WBV studies and we determined that we were willing to collaborate with the aim to update/develop reporting guidelines.2.Pre-work by a small group of experts. A small group of five experts had face-to-face meetings and studied the literature for existing WBV reporting guidelines, guidelines for connected topics, and the use of these guidelines. We decided that there is a need to update and expand the guidelines from Rauch et al. [24].3.Executive group. We established the final executive group, including 15 experts from Brazil, Germany, France, Spain, Portugal, Hungary, the USA, and the Netherlands. This executive group promotes interdisciplinary collaboration with the members’ backgrounds in (bio)engineering, neurobiology, medicine, physiotherapy, psychology, and human movement science.4.Registration. We registered our reporting guidelines under development in EQUATOR [29].5.Delphi studies. We performed two Delphi studies. The first Delphi study was for WBV research in human populations, and the second, a small additional Delphi study, was for WBV in animals and cell cultures. The Delphi study for human populations used the existing guidelines of Rauch et al. [24] as a starting point. This Delphi study is published elsewhere [30]. In total, 51 expert researchers from 17 countries with, on average, over 19 years of WBV research experience participated in this Delphi study, which was conducted in three rounds. The identified list of potential topics formed the basis of the updated and developed reporting guidelines. More detailed information can be found in Wuestefeld et al. [30]. The additional Delphi study for WBV in animals and cell cultures used the human Delphi study as a starting point. For the outcomes of this Delphi study, we refer to the Appendix A (Appendix A).6.Discussion of the outcomes of the Delphi studies. We planned a face-to-face meeting with the executive group during the scheduled WAVEX conference in Cologne (2020) to discuss the outcomes of the Delphi studies. However, due to the COVID-19 pandemic, this conference was postponed as travel possibilities were limited. Therefore, we decided to discuss the outcomes in writing and with online meetings.7.Final list of topics and writing the document. The executive group established the final list of checklist items and formulated the connected explanations and elaborations.

## 3. The Checklists

In Table 1, the checklist of information to include in human WBV studies is presented and, similarly, in Table 2, the checklist for animal and cell cultures studies. We define a cell culture either as cells or tissue removed from an animal/human or a cell line/cell strain. They are subsequently grown in a favorable artificial environment containing fresh growth medium. For both checklists, the items are structured into five categories: device, vibrations, administration, general protocol, and subjects.

## 4. Explanation and Elaboration Checklist for WBV Studies in Humans

### 4.1. Information about the Device


**Item 1: For commercial devices: the manufacturer and production type; for commercial and non-commercial (self-built) devices: device specifications**


Explanation

The following information should be given:For commercial devices: the manufacturer (name, city and country) and production type; for non-commercial (self-built) devices: a sketch of the system, and the model number used (so that authors can have a record of which model was used in each study);Size of platform;If applicable, other dimensions of the device;Source of vibration;How the plate is attached to the source of vibration (for example, at the center of the plate/cage or at each corner of the cage/plate).

Examples


1.“The intervention was performed on a <type> standing vibration platform (manufacturer), which has been described elsewhere <reference>. Briefly, the device is x cm * y cm * z cm large, has x kg mass, and can deliver sinusoidal (deviation <5%) vibration through eccentric rotation that is transmitted to the foot plate through a 2-sided driving rod” [31]. The device can produce vibrations with peak-to-peak displacement up to 8 mm within a frequency range of 8–30 Hz.2.“Subjects were tested with a custom-made vibration platform that uses a combination of eccentric rotation with springs attached to a base plate and foot plate” [31] (see Figure x). Specifically, we used engineering model #4 in this study. This device is x cm long, y cm wide, and z cm high and has a mass of x kg. It delivers a constant non-loaded f Hz vibration that is near sinusoidal (x% deviation, see spectrogram in Figure y) with a non-loaded peak-to-peak displacement of z mm. When loaded with x kg, the vibration frequency is f Hz, and the peak-to-peak displacement is x mm.



**Item 2: If applicable: platform constructions (e.g., mounting a chair on it)**


Explanation

Information should be given where and how the platform was placed and whether any additional structures were mounted; where necessary, provide a figure.

Examples


1.The vibration platform was placed on the ground, and a handrail was mounted to facilitate balance control during deep squats.2.“To this purpose, a prototype vibration machine consisting of a motorized horizontal leg press with a pin-weighted plate was fitted with two electrically powered 0.15-kW vibration actuators <model, manufacturer, city> to the rear of the footplate of the leg press machine (Figure x)” [32].


### 4.2. Information about the Vibration


**Item 3: The type of vibration: spatial and temporal characteristics (e.g., vertical or side-alternating)**


Explanation

Vibration platforms can be designed such that the resulting accelerations are delivered in a single direction (e.g., vertically only) within a 2D plane defined by two directions (e.g., within the horizontal plane without a vertical component) or in any direction within a 3D space. Further, the direction of the acceleration can be time-variant or time-invariant. For side alternating vibration platforms, the direction of the displacement/acceleration vector changes throughout a sinusoidal cycle and is dependent on the spatial location of the point on the vibrating plate where it is measured. For synchronous vibrations (i.e., all spatial locations of the vibrating plate experience the same vibration characteristics), the displacement vector can either consistently point into a given direction (e.g., horizontal or vertical) or change temporally in direction in 2D or 3D. Lastly, the waveform of the vibration produced by the platform should be measured and characterized. Examples for potential waveforms include, but are not limited to, sinusoidal (most common waveform), trapezoidal, sawtooth, and hammer forms.

Examples


1.Recordings from accelerometers confirmed that the platform produced sinusoidal accelerations with a predominant vertical component and much smaller horizontal components (rms acceleration in the horizontal direction were less than 10% of those in the vertical direction). The direction of the produced acceleration (displacement) did not change significantly over any given sinusoidal cycle.2.The side-alternating vibration platform produced sinusoidal vibrations within a plane defined by the vertical and medio-lateral directions. Inherently, the direction of the accelerations was time-variant.



**Item 4: The vibration parameters: definitions and parameter settings used**


Explanation

Because of the inconsistent use of terminology [22], it is important to report definitions of the vibration parameters and/or refer to papers in which the vibration parameters are defined clearly (e.g., current paper, Rittweger [19], and Rauch et al. [24]). Specifically, reporting the magnitude of the vibrations is prone to inadequate use since it can be expressed in different units related to displacement or acceleration. Moreover, the wave form may deviate from pure sinusoidal information (as described above).

We recommend the following parameter definitions:


Frequency: number of cycles per second (in Hz);Peak-to-peak displacement: distance between the minimal and maximal positions of the platform or device (in mm);Peak accelerations: maximal and minimal acceleration within a cycle (in multiples of Earth’s gravity g); not to be confused with peak-to-peak displacement;rms acceleration: root-mean-square acceleration (in multiples of Earth’s gravity g);Duration: duration of one bout of WBV (either in minutes or seconds);Number of bouts: number of WBV periods that are alternated with periods of rests within one session;Rest period: If a multiple number of bouts is applied: the duration of the rest period between bouts (in minutes or seconds).


For a description of vibration magnitude, there are three parameters included, namely, peak-to-peak displacement, peak accelerations (maximal and minimal), and rms acceleration. Full 3D information should be obtained, and the results of all three components in the three spatial directions should be used for reporting magnitude parameters.

Whilst a pure sinusoidal oscillation can be sufficiently described with one magnitude parameter only (Figure 2), all available vibration machines generate oscillations that, under real-world conditions, depict small or even large deviations from sinusoidal waveforms. Thereby, the use of three magnitude parameters allows us to gauge such deviations from sinusoidal waveforms.

Note: We explicitly discourage reporting displacement amplitude, as it is redundant with peak-to-peak displacement in sinusoidal oscillations and not defined for non-sinusoidal oscillations.

Note also: For sinusoidal oscillations, peak-to-peak displacement, peak acceleration, and rms acceleration are mathematically related, which is not the case for non-sinusoidal oscillations (Figure 2).

Example

In accordance with <current paper>, the vibration frequency (Hz) is given as the number of cycles per second. The vibration magnitude is expressed as peak-to-peak displacement (D_peak-to-peak_, in mm, defined as the distance between minimal and maximal vertical positions of the platform), peak acceleration (a_peak_, in g, defined as maximal vertical acceleration within a cycle), and root-mean-square acceleration (a_rms_, in g). During the intervention, a vibration frequency of xx Hz was used, and the subjects’ feet received D_peak-to-peak_ of x mm, a_peak_ of y g, and a_rms_ of z g.


**Item 5: Whether and how the vibration parameters were verified**


Explanation

Vibration parameters often differ from manufacturer specifications, and the trueness of parameters will change during any device’s lifetime [33]. Hence, in addition to the manufacturer’s statements, vibration parameters have to be regularly assessed in practice. High-quality studies, therefore, should report how the vibration parameters were verified. If this is not possible, then the manufacturers’ values should be reported and stated as such. This is straightforward to do via accelerometers that are attached at the representative spot(s). Alternatively, one could also assess the displacements with laser distance measurements. In any case, the measurement device should capture frequency components of 500 Hz or higher. Measurements should not only be taken with the unloaded vibration device but foremost while a person is performing the interventional exercises. Since many vibration devices do not provide pure sinusoidal vibration, a set of parameters has to be derived [34,35,36,37], including but not limited to peak-to-peak displacement, peak acceleration, and rms acceleration [38]. In addition, it is recommended to provide a spectrogram and to give an estimate for the deviation from a sinusoidal shape. This is important as higher frequency components (e.g., impacts) have been accused of being detrimental to health [39]. It should also be considered that vibration parameters can vary with the user’s body mass and the type of ground matter (e.g., cement vs. wood flooring vs. carpet); they can also vary with time due to the wear of the device. The use of low/high pass filters in data analyses for attenuation noise may be reported depending on the nature of the study and the technical detail required.

In summary, it is preferred to verify the vibration parameters, as given by the manufacturer, with actual measurements under different conditions (with and without the subject), and sufficient detail about the measurements and subsequent data analyses should be reported.

Example

Trueness of vibration parameters was assessed using a <type> 1D accelerometer (<manufacturer>). This accelerometer deviates from linearity by <3% up to 1000 Hz. The accelerometer was firmly affixed to the platform by a bolted cage, and recordings were taken close to the foot of subjects while performing squatting and calf-raising exercises. The results yielded that the true vibration frequency was f Hz (SD f.SD), peak-to-peak displacement was x mm (SD x.SD), rms acceleration y mm (SD y.SD), and peak acceleration z g (1 g = Earth’s gravitational acceleration, SD z.SD). These parameters were constant throughout the study (*p* > 0.20) and did not vary with the subjects’ body mass (*p* > 0.20).


**Item 6: For side-alternating vibrations: the location on the vibration platform where the accelerometer was placed to measure magnitude**


Explanation

For quantifying vibration characteristics of side-alternating vibration platforms, it is imperative that the location(s) at which the accelerometer(s) is (are) placed is described precisely. Inherently, the farther the distance between the position of the accelerometer and the pivot, the greater the measured acceleration/displacement. Ideally, the accelerometer should be positioned at the sites where the subjects will be standing on the platform. If that is not possible, it should be stated where subjects placed their feet and what the expected peak-to-peak displacement is for exposure of the subjects.

Example

3D accelerometers were placed at x cm and y cm distance from the pivot on either side of the plate. Accelerometers were centered fore-aft. Accelerations and displacements were determined as the root-mean-square (rms) of the vertical and horizontal components. Vibration magnitude at the accelerometric measurement site amounted to x for peak-to-peak displacement (D_peak-to-peak_), y for peak acceleration (a_peak_), and z for rms acceleration (a_rms_). The measured a_peak_ and a_rms_ values deviated by x% and y%, respectively, from a sinusoidal oscillation with frequency z1 and D_peak-to-peak_ z2. […] During the intervention, subjects placed their feet x cm away from the platform’s rotation axis, thus perceiving a D_peak-to-peak_ of x’ mm, an a_peak_ of y’ g, and an a_rms_ of z’ g.


**Item 7: Whether frequency and magnitude were constant or modulated**


Explanation

Most studies use a constant frequency and magnitude, but others have allowed the modulation of these parameters in any manner (e.g., random, increasing). That elicits continuous or rapid alterations of vibration frequency and/or magnitude. It is therefore important to report whether such modulations occurred and how they affected the vibration parameters. This also includes any parameter alterations during switching on/turning off if that affected subject exposure. If the vibration was meant to be constant, then an estimate of the variation within a typical session should be given, e.g., as standard deviation or variation coefficient.

Examples


1.The vibration platform provided a constant vibration, with SD of x Hz for frequency of y mm for D_peak-to-peak_, z mm, z g for a_peak_, and zz g for a_rms_, as confirmed by accelerometers attached to the plate. The device was always switched on before subjects stood on it, and they stepped off before the device was turned off.2.The vibration magnitude was modulated by a sine function, resulting in a variation of D_peak-to-peak_ from x to y mm, a variation in a_peak_ between x’ and y’ g, and a variation in a_rms_ between x’’ and y’’ g.


### 4.3. Information about the Administration


**Item 8: The posture or body position of the subject and whether it was changed during the intervention (static versus dynamic exercise)**


Explanation

Body position is considered a key parameter to understanding how vibration is transmitted through the human body (i.e., transmissibility) [40]. Numerous studies have analyzed the transmissibility of the vibration to different locations on the body, even in a sitting position [41]. This is also the case during training [40,42], showing that vibration, especially in a vertical direction, might play an important role in the biodynamic response of the spine depending on the posture (e.g., lean forward or backward). Moreover, during dynamic contractions, muscle activity was reported to be greater than in static contractions [43]. The transmissibility to the head also differs with the posture, leading to an increased risk of injury (i.e., resonant frequency). This will be detailed below (Item 11), discussing how vibration transmission to the head is prevented.

Examples


1.“A training session consisted of eight different dynamic and static exercises (lunge, step up and down, squat, calf raises, left and right pivot, shoulder abduction with elastic bands, shoulder abduction with elastic bands while squatting, arm swinging with elastic bands; ESM Appendix). During the isometric squat exercise, the subjects were instructed to stand with bent knees and hips on the platform with a 100° knee flexion (considering 180° as full knee extension). The dynamic exercises were performed with slow movements at a rate of 2 s for both concentric and eccentric phases” [44].2.“A wooden plate (0.5 m × 0.9 m × 0.02 m) was mounted on the vibrating platform to enlarge the platform of the device. In order to apply passive WBV, a chair (with armrests and a seating area of soft material) was firmly mounted on the wooden plate to control the activity and movement of the subjects (Figure 3)” [11].



**Item 9: The position of the feet of the subjects on the platform during the vibration and how the feet were loaded (e.g., on midfoot or on forefoot with heel lifted off)**


Explanation

The position of the feet has also been reported to influence EMG activity and, consequently, the transmission of the vibration [45]. Raising the heel from the vibration platform reduces vibration transmissibility [46]. Heel raises were used as an exercise in different studies (i.e., single-leg heel raises on each leg from maximal plantarflexion to maximal dorsiflexion or double-leg heel raises), leading the vibration energy to focus on the shank muscles [46]. This position was reported to improve muscle strength and power in the calf muscles and, to a lesser extent, the foot dorsiflexor muscles.

Examples


1.“Feet shoulder-wide apart or slightly wider, toes pointing slightly outward; knees almost straight but not locked; raising heels from the ground. The subjects were instructed to move at a pace of 0.5 Hz, i.e., 1 s up on to toes to maximum heel raise and 1 s down to complete flat foot and to ensure each repetition is a full heel raise, i.e., as far up onto their toes as possible” [47].2.Subjects stood on the WBV platform with feet flat on the platform, positioned at shoulder width, and knees approximately at 45° of flexion.



**Item 10: If and how skidding of feet was prevented**


Explanation

Skidding or sliding movements on the vibration platform are a sign of un-firm coupling between user and device. Skidding is, therefore, a mechanism by which impacts can be generated [19]. These are potentially detrimental to health [39]. Accordingly, skidding should be prevented. This is usually possible by more crouched positions [46] and in a standing posture by lifting the heels from the ground.

Example

In n (= x%) of subjects, skidding of the feet occurred during the initial first or second bout. Subjects were then instructed to assume a more crouched position and to raise their heels a little further so that skidding was prevented in all subjects.


**Item 11: If and how vibration transmission to the head was prevented**


Explanation

Mechanical energy transferred to the head can be hazardous, and it is well documented that frequency and knee flexion angles might affect acceleration transmission to the head [48]. As reported in Item 13, certain postures can increase the transmissibility to the head. Flexing the knees has been found to decrease head accelerations using both vertical synchronous WBV [49] and side-alternating WBV [50], but again, depending on the frequency used. Despite some contradictory results reported, Abercromby et al. [51], using a fixed configuration (30 Hz and 4 mm), suggested that head acceleration seems to decrease when knee flexion angles are increased from 10° to 30°; however, the authors suggested an increase an in transmissibility above 30° of knee flexion. These results contrast with previous studies suggesting that flexing the knees results in reduced head transmissibility at all frequencies [48]. It seems that the transmission of vibration to the head is reduced with angles greater than 20° [48]; in fact, it was recommended to avoid frequencies below 30 Hz, with small knee flexion angles (<40°) to reduce the risk of injury at the head level [48]. These suggestions can also be applied to WBV using dynamic movements. In a recent study, Caryn and Dickey [48] assessed transmissibility to the head (frequencies between 20 and 50 Hz) during dynamic squats and showed that the response to dynamic exercise is similar to static postures. Hence, summarized, vibration frequency and knee flexion angle interact to affect acceleration submission to the head.

Example

A subject stood in the squat position with 30° knee flexion on the vibration platform for five 1-min bouts of vibration with 1-min rest between each bout. Peak-to-peak displacement and frequency were set at 4.5 mm and 30 Hz, respectively, so as to produce the highest acceleration. The accelerometer was set to 100 Hz and recorded the resultant acceleration for each 1-min bout. On a second occasion, the accelerometer was gripped between the teeth in order to measure the acceleration experienced at the head.


**Item 12: If a handrail was available/used**


Explanation

Individuals using WBV platforms may be at risk for falls due to imbalance and/or disorientation. Some WBV platforms provide safety measures, such as handrails, to prevent these issues. However, depending on the population or application, it may be allowed or not to use the handrails, and this should be mentioned.

Examples


1.Subjects were instructed to stand on the WBV platform, not touching the handrail. Trials were repeated if subjects touched the handrail for support.2.Subjects were instructed to gently hold the handrail to prevent falls.



**Item 13: The position of the hands during the WBV and if the hands were directly subjected to the vibration**


Explanation

The position of the hands determines whether the hands were directly subjected to the vibration. Therefore, the position of the hands must be described.

Examples


1.Subjects were instructed to stand on the WBV platform, with head and eyes facing forward, body mass distributed on both feet, and arms outstretched and palms facing down.2.Subjects were instructed to stand on the WBV platform, with head and eyes facing forward, weight distributed on both feet, and arms outstretched and grasping the platform’s handrail.



**Item 14: If applicable, the parts of the subjects’ body that were most subjected to vibration (e.g., predominantly the feet)**


Explanation

The vibration is mostly transmitted to body parts placed in contact with the vibration platform, as reported by objective [52] and subjective [53] measures. The aim of the study will determine the position and, consequently, which body part is directly subjected to vibration. The rationale behind the decision of which body parts (e.g., predominantly the feet) of the subjects were most subjected to vibration should be provided.

Examples


1.Subjects were positioned in the push-up position on the vibration platform, with their hands apart at 30 cm, directly on the platform, to maximize the transmission of vibration to the forearm muscles.2.The squat exercise was performed on a side-alternating platform. The side-alternating movement of the platform evokes muscle contractions on the entire flexor and extensor chain of muscles in the legs.



**Item 15: Whether and, if applicable, what tools/aids were used during the vibration (e.g., type and size of dumbbells or resistance bands)**


Explanation

Additional loads superimposed to WBV can cause higher neuromuscular responses [54] and have been reported to have a beneficial effect on performance and metabolic power [55,56]. Ritzmann et al. [54] assessed the influence of different WBV determinants on electromyographic (EMG) activity using one-third of body weight as an additional load (via a bar on the subjects’ shoulders). Authors reported increased EMG activity in the *M. soleus*. Similarly, Hazell, Kenno, and Jakobi [57] reported an increase in EMG activity during WBV with an additional load in a dynamic squatting condition. Therefore, considering that an external load can determine the effectiveness of WBV exposure, authors should report the use of tools/aids used during vibration.

Examples


1.“Upper body exercises included static biceps curls and triceps extensions. These were performed by holding nylon straps attached directly to the surface of the WBV platform at the peak-to-peak displacement of 8 mm and frequency of 40 Hz” [58].2.“Vibration was with a peak-to-peak displacement of 11 mm, a frequency of 26 Hz, and, hence, a peak acceleration of 147 m.s^−2^, or 15 g. The subjects bore an additional load fixed around the waist (40% of body weight in males; 35% in females because of their higher total body fat mass). After 30 s of simple standing, they started squatting, i.e., bending their knees in a 6 s cycle, 3 s down and 3 s up, as smoothly as possible” [59].



**Item 16: General exercise parameters (e.g., duration, number of bouts, rest intervals)**


Explanation

Generally, exercise is described by parameters such as mode, frequency, intensity, duration, and activity-rest pattern [60,61]. In the context of WBV exercise, the mode may refer to whether it is imposed to active subjects (performing exercises) or more passive subjects (holding posture, sitting, or lying) (see Item 8). The frequency of WBV sessions is expressed in the number of sessions per week and/or per day. The intensity is determined by the vibration parameters (see Item 4, not mentioned further here) but can also be determined by other factors (e.g., additional exercises performed during WBV (Item 8)). The intensity of these additional exercises can be expressed in terms of absolute intensity (e.g., back squatting with 60 kg) or relative intensity (e.g., back squatting at 70% of 1 RM). The duration may apply to bout, session, or program duration and is mostly expressed in minutes (bout and session duration) or weeks (program duration). The pattern reflects the temporal aspects of bouts of exercise and rest intervals. Finally, for WBV programs/interventions, attendance should be reported, defined as the percentage of WBV sessions attended out of the actual number of sessions offered [62]. Table 3 gives an overview of the general exercise parameters relevant for WBV.

Examples


1.WBV intervention program: The subjects performed WBV exercises twice a day, 5 days per week for 8 weeks.2.One session of WBV: Each WBV session consisted of three bouts of WBV, with a duration of 2 min and 3 min of rest between the bouts. During the WBV exercise, the subjects performed squats without additional weight (first bout) and then with an additional load using barbells at 20% of 1RM (second bout) and 40% of 1RM (third bout).


### 4.4. Information in General Protocol


**Item 17: The setting of the WBV sessions/intervention (e.g., hospital, gym, or at home) and the time of day the sessions took place**


Explanation

WBV sessions may be performed in different settings (where) and at different times of the day (when). It is important to report the setting and time for reproducibility reasons. Typical settings include, but are not limited to, hospitals, rehabilitation units, research laboratories, gyms, and subjects’ homes. Irrespective of the setting, to understand the environmental conditions, it is important to report the specificities of the location (e.g., hospital ward, individual therapy room, common rehabilitation area, research laboratory, common training area, living room), if the training was individual or in a group, and whether there was something the subject could hold on to (e.g., a table). In addition, at what time of day the sessions took place should be reported, e.g., in the morning, afternoon, or evening, being specified with precise times or in relation to the circadian rhythm.

Examples


1.The WBV sessions took place in a gym, in a common training room in which other sportspeople were training at the same time. All sessions took place between 8.00–10.00 p.m.2.The WBV sessions took place at the homes of the subjects. Generally, the device was placed in the living room or a bedroom with at least 0.5 m of free space around the device.3.The training sessions were carried out individually in the research laboratory of <insert institution>.



**Item 18: Presence of a trainer (face to face/online) during the WBV sessions**


Explanation

The presence of a qualified trainer or therapist may facilitate the correct performance of the vibration exercise, assure safety, and promote attendance. In general, supervised (vs. unsupervised exercise) promotes higher attendance [63] and accentuates exercise effects in several populations and outcome domains [64,65]. More specifically, unsupervised WBV is discouraged because of higher dropout rates and lower efficacy [66]. To interpret the outcomes of WBV studies correctly, it is important to report the presence or absence of a trainer or therapist during the WBV sessions and if the presence was face-to-face, online, or otherwise. Furthermore, it is important to report if the WBV sessions were completely or partly supervised by a trainer or therapist.

Examples


1.All WBV sessions were supervised online using <name tool> by an in WBV experienced exercise trainer.2.The first 6 WBV sessions (33.3%) were supervised by a physical therapist, and the remaining 12 WBV sessions were unsupervised.



**Item 19: The instructions given to the subject before the WBV session**


Explanation

It should be reported how, when, and what instructions were given to the subject. How instructions were given may refer to face-to-face contact, an online video session, an online training manual, and/or a printed training manual. When instructions were given refers to the time between the instructions and the start of the WBV exercises (e.g., one week before the first WBV session and/or at the start of each WBV session). What instructions were given includes instructions regarding clothing and shoes, intake (food, drinks, medication) before the session, other exercises prior to the WBV session, safety instructions, and instructions about the performance of the WBV exercise. For the latter, we refer to the items about administration (Items 8–16).

Examples


1.One week before the WBV session, the subjects received an information letter by email in which they were instructed to avoid the intake of stimulants such as coffee, alcohol, or drugs for at least 12 h before the session, to maintain normal medication, and to wear comfortable clothing during the WBV session.2.At the beginning of the WBV session, the supervisor explained the posture/exercises to be executed during the WBV session.



**Item 20: Preparatory exercises or warm-up prior to the vibration (type and duration of exercises)**


Explanation

Information regarding the warm-up should include the following: (1) time (e.g., 5 min), (2) type (e.g., cycle ergometer), and (3) time between the warm-up and the main exercise (e.g., after 2 min, subjects performed 6 sets of 1-min WBV).

Examples


1.Warm-up using WBV: The individuals performed 30 s of the WBV intervention as a warm-up modality before the experimental session in all sessions. The frequency used in this warm-up was 30 Hz, 2 mm of peak-to-peak displacement, barefoot, with 130° of knee flexion, using a vertical platform.2.Warm-up without WBV: subjects performed a standardized warming-up protocol consisting of 5 min of jogging at a self-selected easy pace, 5 min of joint mobilization exercises, 10 squats without an external load, 5 countermovement jumps, progressive in intensity, with 1-min rest periods between them.



**Item 21: The subjects’ footwear (shoes, socks, barefoot) during the vibration with a detailed description**


Explanation

Marín et al. [67] assessed the effects of footwear during WBV and suggested that athletic shoes could alter part of the vertical vibration. They showed that the muscle activity of the vastus lateralis or the gastrocnemius medialis was different according to the type of shoes or without shoes. These changes were attributable to the dampening of the vibration with pliable soles or even the increased surface area of the foot that is in contact with the platform surface, especially with running shoes. Information on the type of sports shoes and interface with the vibration platform is crucial to judge the transmissibility of the vibration. Authors should report the use or not (e.g., barefoot) of shoes, the type of sole and midsole, if possible, and interface with the vibration platform [68,69].

Examples


1.“Women in the platform group stood still and barefoot on the platform for 20 min, 5 times a week for 12 months” [70].2.“The footwear that was worn during the study included running shoes, basketball shoes, and tennis shoes” [67].3.“Volunteers were barefoot to prevent the attenuation of vibration by shoes. To reduce the risk of sliding, non-slip foam of <number> mm was attached to the platform surface” [68,69].



**Item 22: If applicable: characteristics control/sham condition or intervention**


Explanation

In the case that a control/sham condition or intervention is part of the study design, detailed information about the nature of the condition and specifications for the actual WBV treatment performed (e.g., setting, instructions, parameters) must be provided.

Examples


1.For both the WBV group and the sham group, subjects stood upright on a <type> vibration platform (manufacturer) with <describe position>. The WBV group received <describe intervention>. The sham group received an intervention following exactly the same procedure as the WBV group except that no “real” vibrations were applied to subjects. To provide the most realistic placebo conditions as possible, subjects were told by the supervisor that they could not physically feel the vibrations because they were too small to be consciously sensed; a device emitting a similar sound to the real vibration platform was used.2.For both the WBV group and the control group, subjects stood upright on a <type> vibration platform (manufacturer) for x minutes with <describe position>. The WBV group received <describe intervention>. The control group received an intervention following exactly the same procedure as the WBV group except that no vibrations were applied to the subjects.3.The WBV group received whole-body vibration exercise <number> times a week for <number> weeks. Each session included <number>-minute warm-up, <number>-minute whole-body vibration exercise, and <number>-minute cool down. The whole-body vibration exercise protocol comprised the following movements: x, y, z <describe>. The control group received general exercise <number> times a week for <number> weeks. Each session included <number>-minute warm-up, <number>-minute exercise, and <number>-minute cool down. The general exercise protocol comprised the following movements: x, y, z <describe>.



**Item 23: The moment at which the outcome measures were assessed: during, before, and/or after the vibration; the time between begin/end of exposure/exercise/session and assessment(s)**


Explanation

In general, outcome measures can be applied before WBV administration, during WBV administration, and/or after WBV administration. Therefore, the time of assessment of outcome measures needs to be defined with as much precision as possible. Additionally, the time periods between WBV administration and outcome measurement (either before or after WBV administration) need to be reported. This is relevant for all kinds of studies, irrespective of if they are focusing on acute effects (i.e., effects measured immediately or soon after WBV administration) or more chronic effects of WBV (i.e., effects measured later in time, e.g., days, weeks, or months after WBV administration has been terminated).

Examples


1.“Each trial started with a two-minute period of experimental treatment, either a period of vibration (vibration condition) or a resting period of no vibration (resting condition). Immediately after the experimental treatment, the Color-Word Interference Test and the Color Block Test of the Stroop Color-Word Interference task were performed. The Color-Word Interference Test always preceded the Color Block Test. Subsequently, a resting period of three minutes followed, prior to the beginning of the subsequent experimental trial” [12].2.“Each trial started with a three-minute period that was either (A) a period of WBV (vibration condition) or (B) a resting period of no vibration (non-vibration condition). Immediately after the experimental treatment, first, the Color-Word Test and then the Color-Block Test was applied. Each trial finished with a break of three minutes” [71].3.“The first assessment was performed on day 1. WBV treatment sessions were applied between day 2 and day 11 for 10 consecutive days (30 WBV treatment sessions in total). The second assessment was performed on day 12 (about 16 h after the final treatment session was completed). Finally, a third assessment (follow-up assessment) was performed on day 25. No treatment or other meetings related to the research project were carried out between the second assessment and the follow-up assessment. All three assessments were performed at the same time of the day (starting at 9 a.m.)” [13].


### 4.5. Information about the Subjects


**Item 24: General characteristics of the subjects**


Explanation

General background characteristics of subjects are important to report in order to judge generalizability, to give information about subject screening, or to compare subgroups, for instance, in reviews and meta-analyses.

General background characteristics of subjects relevant for WBV research may include:Age and sex;Body height, body mass, and, if possible, body composition (percentage body fat);Anthropometric measurements (e.g., waist circumference, hip circumference, and neck circumference);Level of physical activity (sports and exercise; if applicable: activity type, frequency, intensity, and duration);History of injuries;If applicable: (co)morbidity;If applicable: the time of disease development/disease progression;If applicable: number and type of medications used;If applicable: (absence of) pregnancy;If applicable: mental and/or cognitive status (e.g., depressive symptoms, global cognitive function).

Examples

For an example of how to put this information in text or a table, we refer to general reporting guidelines, e.g., the Publication Manual of the American Psychological Association [72] or specific journal submission guidelines for authors.


**Item 25: The subjects’ previous experience with WBV**


Explanation

Subjects’ previous experience with WBV should be reported for several reasons. First, the physiological responses to WBV may depend on the amount and quality of prior experience [73]. Furthermore, novelty effects may confound results in subjects without any experience with WBV, especially if outcome measures are in the field of subjective experiences measured with questionnaires. Finally, the progression principle of exercise may apply, which may affect the individual’s optimal vibration parameters [74].

The subjects’ previous experience with WBV may include:Amount of previous experience;How long-ago previous experience happened;The setting of previous experience (e.g., at home, in a gym, in a rehabilitation center);If previous WBV was supervised by an experienced trainer or therapist;If known, device, parameters, and positions used in previous experience.

Examples


1.None of the subjects had any previous experience with WBV prior to participating in the experiment.2.One subject in the experimental group had previous experience with WBV. Three years ago, in a gym, he performed WBV exercises in a squat position for about 6 sessions using a side-alternating vibration platform (further details unknown).



**Item 26: Acute, short term or long-term side effects of the vibration exercise**


Explanation

In general, all kinds of side effects may occur following WBV, whether they be positive, neutral, or negative in nature. While one side effect may be regarded as positive by one person but negative by another person, it is paramount to inform about all side effects that are reported by the subjects. As side effects may occur during WBV exposure, immediately after, or after a delayed time period, the time point of the occurrence of side effects also needs to be reported. It is desirable to provide detailed information about the time course and intensity of side effects.

Examples


1.After the WBV session, six subjects reported physical sensations of short duration such as muscle trembling, a tingling feeling, or weak knees. One subject experienced the session as uncomfortable, and one subject reported a mild headache, which disappeared after half an hour. There were no serious adverse side effects.2.“After reading the information letter and signing the informed consent form, the subjects were asked to fill in the questionnaire for the first time (pre-test questionnaire). (…) After finishing the pre-test questionnaire, subjects were asked to sit on the wooden chair on the WBV-device, with their feet on the platform, their arms on the armrests, and their back on the backrests of the chair. Immediately after the WBV session, subjects were asked to fill in the questionnaire for the second time (post-test questionnaire). Finally, subjects were asked to fill in the questionnaire again via a link sent by email, 24 h after the WBV session (follow-up questionnaire)” [75]. For a forest plot of how 88 subjects felt immediately after WBV (intensity of experiences) in effect sizes (Cohen’s d) and 95% CI’s (adopted from Oerlemans et al. [75]), see Figure 4.


## 5. Explanation and Elaboration checklist for WBV studies in animals and cell cultures

### 5.1. Information about the Device


**Item 1: For commercial devices: the manufacturer and production type; for commercial and non-commercial (self-built) devices: device specifications**


For explanation and elaboration, see Table 1, item 1.


**Item 2: The dimensions of the cage or container in which subject(s) was (were) placed during WBV**


Explanation

If animals are placed in a cage, box, or container that is placed on or attached to the vibrating plate, it is important to report the dimensions (height, width, and depth in centimeters). Based on this, the reader can judge to what degree the animals can freely move. In addition, was a lid placed on the cage or box to prevent the animals from escaping? Preferably, a picture of the vibration platform, including the cage(s) with the animals or cell cultures, should be depicted in a figure.

Examples


1.The animals were placed in a cage (20 × 20 × 30 cm) in which they could move around freely. A transparent lid was placed on the cage to prevent the animals from escaping.2.The animals were placed in small boxes (15 × 10 × 10 cm), which limited their possibility to move around but were large enough to prevent substantial immobilization stress. No lid was placed on the boxes because the animals did not make any attempt to escape.


### 5.2. Information about the Vibration


**Item 3: The type of vibration: spatial and temporal characteristics (e.g., vertical or side-alternating)**


For explanation and elaboration, see Table 1, item 3.


**Item 4: The vibration parameters: definitions and parameter settings used**


For explanation and elaboration, see Table 1, item 4.


**Item 5: Whether and how the vibration parameters were verified**


For explanation and elaboration, see Table 1, item 5.


**Item 6: For side-alternating vibrations: the location on the vibration platform where the accelerometer was placed to measure magnitude**


For explanation and elaboration, see Table 1, item 6.


**Item 7: Whether frequency and magnitude were constant or modulated**


For explanation and elaboration, Table 1, item 7.

### 5.3. Information about the Administration


**Item 8: Whether the animals were housed individually or in groups during the WBV session**


Explanation

If animals are placed in a cage, box, or container that is placed on or attached to the vibrating plate, it is important to report whether this deviates from their normal housing situation. If animals are group-housed but individually placed on the vibration platform, the absence of an individual from the group could influence social hierarchy. Placing the animals back in the group after a WBV session may, therefore, induce social stress (e.g., fighting, as sometimes seen between male mice). If the animals are routinely housed individually but placed on the vibration platform as a group, this may induce considerable social stress during the vibration sessions. Social conflicts can cause physical and mental stress, potentially affecting certain outcome parameters of a study.

Examples


1.Animals were placed as a group on the vibration platform. The composition of these groups was identical to the groups in which they were housed during the entire experiment.2.Animals were individually placed on the vibration platform. After the session, the animal was placed back in the group in which it was housed during the entire experiment. Special attention was paid to determine any signs of social stress or changed social interaction between the animals due to the temporary change in the composition of the group.



**Item 9: Whether the animals were habituated to the device before the start of the WBV protocol**


Explanation

If animals are placed in a cage, box, or container that is placed on or attached to the vibrating plate, it is important to know whether they have been familiarized with this environment before the actual vibration session(s) start. Any novel environment or a change in physical activity (either enhanced or reduced) can induce novelty stress. By habituating the animals to this new environment, novelty stress can be reduced or even prevented. Novelty stress can induce anxiety-related behavior and alter the physical and mental conditions of the animal, potentially affecting certain outcome parameters of the study.

Examples


Before the actual vibration sessions started, the animals were habituated to the platform. They were placed on it for 10 min 2 or 3 times on the day prior to the start of the experiment. During the habituation, the animals were monitored and their general activity was recorded. Only the animals that still showed signs of novelty stress (increased defecation and urination; frequent rearing behavior) after the second habituation received a third habituation session.Animals were not habituated to the platform prior to the start of the experiment because this strain is known for its low level of novelty stress.



**Item 10: Whether the animals were constrained or not during WBV**


Explanation

If animals are physically constrained while on the vibration platform (other than by limited space in a cage or box), it is important to report the procedure. If, for example, the animals are taped to the vibrating plate, immobilization stress is evident. A clear rationale for doing so should be provided. Immobilization can cause physical and mental stress, potentially affecting certain outcome parameters of the study.

Examples


1.The animals were placed in a plastic tube (give dimensions) in order to block their movements during the vibration sessions. Hence, any form of physical exercise was prevented this way.2.The animals were briefly anesthetized and subsequently taped to the vibration platform to prevent the animals from moving and to induce the required immobility stress (for example, if the aim of the study is to examine the impact of whole-body vibration on immobility stress).



**Item 11: Whether the animals were anesthetized or not during the WBV session**


Explanation

If animals are anesthetized while on the vibration platform, it is important to report the anesthesia procedure in detail. If, for example, the animals are taped to the vibrating plate, immobilization stress is evident. Were the animals anesthetized during the entire vibration session or not? How long did the anesthesia last after the WBV session? A clear rationale for using anesthesia should be provided as well. Anesthesia can cause physical and mental stress, potentially affecting certain outcome parameters of the study.

Examples


1.The animals were anesthetized, taped to the vibration platform, and remained anesthetized during the entire whole-body vibration session to ensure the required immobility. Animals were anesthetized over a period of 30 min per day for 10 days in total.2.The animals were briefly anesthetized and subsequently taped to the vibration platform to prevent the animals from moving during taping.



**Item 12: Wat posture or body position the animals took on during the vibration (e.g., sitting, standing, lying, depending on the species used)**


Explanation

The body position of the animal during vibration is important to judge which body parts receive most of the vibrations. It indicates whether front or hind legs are important targets, to what degree whiskers (if present) are vibrated, and to judge whether all parts of the body receive the vibration to a comparable degree.

Examples

1.Typically, the animals were standing on their hind legs during vibration.2.Typically, the animals walked around in a relaxed way during vibration.


**Item 13: Whether the position/posture/motor behavior of the animals changed during WBV**


Explanation

The behavior, next to the body position, of the animal during vibration can be indicative for the interpretation of how the animals perceive the vibration. Are they aroused, relaxed, or do they fall asleep, for example? Reporting the behavior of the animals during vibration is, therefore, of relevance. It can also change over the course of a vibration session or vibration sessions. For example, in a session of 10 min, a certain type of behavior/body position is observed most frequently during the first minutes, whereas another type of behavior/body position dominates during the last minutes of a vibration session. The behavior may also be different across multiple WBV sessions.

Examples

1.Animals were monitored during the vibration session. Typically, the animals showed frequent rearing behavior, standing on the hind legs, at the beginning of the session. Thereafter, this behavior declined, and, instead, animals started to lie down on the vibration platform. Most of them fell asleep at the end of the session.2.Animals were monitored during the vibration session. No differences in overt behavior or body position were seen within a session or between sessions. The animals typically walked around in a relaxed way.


**Item 14: Whether other sensory systems were stimulated during the vibration**


Explanation

Usually, vibration platforms also produce sound. Hence, next to vibrations, animals are also exposed to sound or even heat produced by the platform. Hence, a vibration session could therefore unintentionally stimulate additional sensory systems. If (one of) the outcome parameters relate to physiology or behavior, it is important that the conditions for all sensory systems are reported. In addition, it is important that the control group, notably the pseudo-vibration group, is exposed to the same sensory conditions as the vibration group.

Examples

1.The noise produced by our vibration platform is around 30 dB. To control for this, the control group (pseudo vibration) was also exposed to this level of noise, in addition to similar handling procedures, as the vibration group.2.Our vibration platform does not produce any noise. The concurrent stimulation of sensory systems other than the one processing vibrations during vibration can be excluded.


**Item 15: General exercise parameters (e.g., duration, number of bouts, rests)**


For explanation and elaboration, see Table 1, item 16.

### 5.4. Information in General Protocol


**Item 16: The location of the intervention (e.g., a separated experimental or test room or the room where the animals were housed)**


Explanation

Animals can be very sensitive to novel environments such as an experimental room during the vibration sessions versus the room where the animals are normally housed, even if they remain in their home cage. If they remain in their home cage during WBV, it is important to report whether bedding was present in the cage as it may absorb part of the vibrations. It is important to report whether the vibration takes place in a separate experimental room. Of note: if the vibration takes place in the room where all mice are housed, and the plate makes noise, it means that all other mice are subjected to the noise as well. The change in the room can come with some mild novelty stress or novelty arousal, which may influence certain outcome measures.

Examples

1.Vibration sessions took place in a separate experimental room. The mice were transported to this room in their home cage 10 min prior to the start of the vibration session.2.Vibration sessions took place in the same room where the animals were housed. Hence, all mice of all groups were exposed to the noise of the vibration platform.3.Animals stayed in their home cage during the WBV session. Cage bedding was removed prior to the WBV session and was put back after the WBV session.


**Item 17: The conditions of the test room (e.g., light and temperature)**


Explanation

In line with point 16, the conditions of the test room could be different from the housing room. These differences can be detected by the animals, triggering a response. These conditions need to be reported in addition to the conditions of the housing room.

Examples

1.Vibration sessions took place in a separate experimental room. Both rooms had the same light conditions and temperature/humidity.2.Vibration sessions took place in a separate experimental room. In this room, the light intensity was reduced, and the temperature was 1 degree Celsius higher compared to the housing room (preferably provide the values for these conditions).


**Item 18: The time of day the WBV session took place (in relation to the light/dark cycle)?**


Explanation

Due to circadian variations/fluctuations in levels of many substances (e.g., hormones, neurotransmitters), it is important to provide information about the time of day the vibration session(s) took place. In the case of multiple vibration sessions, were these given at the same time every day? Or were they spread (randomly) over the day? To put this in the right perspective, the LD cycle should be given, as well as the start of the light phase.

Examples

1.The animals were placed on a 12:12 LD cycle (lights on at 7:00 p.m./lights out at 7:00 a.m.). Vibration sessions took place at fixed times of the day: at 9:00 a.m. and 3:00 p.m. (9:00 and 15:00).2.The animals were placed in constant light conditions. Vibration sessions took place at random times over the day to prevent anticipatory events [76].


**Item 19: If applicable: characteristics control/sham/pseudo condition or intervention**


Explanation

To examine the effects of WBV, it is critical to compare the outcome measures of the experimental group or condition with those of a control/sham/pseudo WBV group or condition. Ideally, all procedures are similar in the experimental and control group or condition, but the actual vibrations only apply to the experimental group or condition.

Examples

1.The pseudo-WBV group was exposed to the same treatment as the experimental group in the absence of actual vibrations.2.The experimental cell cultures were on the vibrating plate and were subjected to vibration, while the control cell cultures were simultaneously placed next to the vibrating plate and did not receive any vibration from the plate.


**Item 20: The moment at which the outcome measures were assessed: during, before, and/or after the vibration; the time between begin/end of exposure/exercise/session and assessment(s)**


For explanation and elaboration, see Table 1, item 23.

### 5.5. Information about the Subjects


**Item 21: General characteristics of the animals/cell cultures**


Explanation

Specific issues that are relevant for vibration detection or responses to vibrations should be mentioned.

Examples

1.The species used in this experiment is known to be highly sensitive to vibrations due to the high density of vibration-detecting receptors located in a specialized organ.2.The type of cells used in this experiment requires low-intensity vibrations to prevent them from entering an inactive metabolic stage.


**Item 22: Whether whiskers were present/intact (if species used have whiskers)**


Explanation

For small rodents, whiskers are important tools for sensory (tactile) information processing. It is likely that the vibrations applied to small rodents are also perceived via the whiskers. Hence, the presence or absence of whiskers (for example, due to “barbering” in the case of group-housed rodents) potentially influences the impact of WBV on certain outcome measurements.

Examples

1.The condition of the whiskers in the animals subjected to vibrations was monitored before each vibration session.2.The whiskers of the animals were trimmed to a standard length to reduce individual variation to the response to vibrations.


**Item 23: The animals’ body weight before and after the entire WBV protocol**


Explanation

The animal’s body weight is an indication of general health. Changes in body weight are, therefore, informative. In addition, if weight is a potential issue in the used subjects (for example, related to obesity, as often seen in otherwise healthy older lab rodents), weight changes become even more relevant.

Examples

1.The body weight of the mice was taken routinely once a week.2.To assess body weight, mice were routinely weighed prior to and 6 h following a vibration session.


**Item 24: Acute, short term, or long-term side effects of the vibrations**


Explanation

If side effects are observed, it is important to determine when these become manifest in relation to the WBV session. The durations of acute, short- and long-term WBV are rather arbitrary. Here, we suggest the following definitions: acute = directly after or during the WBV session, up to 15 min; short-term = after 15 min, up to 3 h; long-term = after 3 h or more. Of note: in principle, a side effect can also be present during all the above-mentioned time periods if the vibration protocol induces a direct and rather permanent side effect.

Examples

1.We checked for any side effects of the applied vibrations, with emphasis on motor functioning and balance, on a daily basis.2.We checked for side effects of the applied vibrations immediately following the vibration session, after 3 h, and after 24 h.

## 6. Discussion

The aim of this study is to evolve guidelines for WBV studies in humans, animals, and cell cultures. Using the EQUATOR methodology, we extended the guidelines of Rauch et al. [24] in multiple ways. First, we incorporated more aspects of WBV research and extended the number of items, especially regarding administration, general protocol, and subjects. In addition, we provided extensive explanations and examples of how to report. Finally, we included adapted and additional items for preclinical research in animals and cell cultures.

The guidelines can be used by researchers, peer reviewers, and editors with the final aim of improving the quality and stimulating the consistency of reporting WBV studies in order to enhance the comparability of outcomes in this research field. Despite this, we want to emphasize that the guidelines are not developed to provide recommendations about the application of WBV. Hence, we do not intend to make statements about preferred devices, parameter settings, or protocols. Furthermore, we do not purport using the guidelines in a rigid way. Not all items will be relevant in every situation, and variations in the way of reporting should be possible. Although the aim is to improve the quality and stimulate consistency of reporting, researchers can use the checklist for designing their studies as well. The items give a broad overview of the relevant aspects of WBV research, which may help to set-up and standardize research procedures and protocols.

The guidelines provide items about WBV-specific aspects. We did not include more general aspects of research, e.g., study design, randomization, blinding, sample size determination, recruitment, outcome measures, statistics, and limitations. For these general aspects, we refer to general guidelines such as the CONSORT statement [26], the STROBE statement [27], or the ARRIVE guidelines [28]. Hence, the guidelines in the current paper should be used in addition to these general guidelines.

In this study, we adhered to the recommendations of the EQUATOR network, and, accordingly, we registered our study as a reporting guideline under development [29]. However, a few remarks regarding our procedures should be made. First, because of the COVID-19 pandemic, face-to-face meetings were not possible from March 2020. Therefore, we discussed the outcomes of the Delphi study in online meetings and in written documents. We do not feel that this adapted procedure affected the content and quality of the guidelines. Further, the outcomes of the Delphi study in humans [30] resulted in a large number of potential items. After comprehensive and in-depth discussions, we reduced the number of items and reformulated items to reduce overlap and promote consistent language. The Delphi study for animal and cell culture studies was based on a limited number of respondents. Although this can be considered as a limitation of this study, we want to stress that insights from the Delphi study in humans are incorporated in the guidelines for animal and cell culture studies. In addition, the international group of authors of the current paper includes extensive knowledge regarding this domain as well, which ensures the quality of the guidelines for animal and cell culture studies. Finally, the last step in the EQUATOR recommendations refers to the publication and dissemination of the guidelines. We propose to add another step regarding monitoring and consolidation. After publication, the executive group will monitor the use of the guidelines and, if necessary, will further disseminate and/or update the guidelines. The regular international WAVEX conferences will facilitate this process.

Throughout the current paper, we used the term whole-body vibration. If applicable, we recommended this term above to, e.g., vibration therapy, vibration exercise, or vibration stimulation. Consistent language facilitates searching in databases for scientific literature and promotes the recognizability of WBV as a therapy or exercise regimen.

We did not propose benchmarks to discriminate between low, moderate, and high vibration frequency and peak-to-peak displacement. Although we see some value for a consensus here, we also realized that such denotations may depend on many factors, including type, direction and shape of the vibrations, population characteristics, and intended outcome measures. Therefore, we recommend reporting the exact and verified values for the vibration parameters without a denotation regarding low, moderate, or high unless specific benchmarks are defined in the paper concerned.

The reporting guidelines were not developed from the perspective of local vibration therapy in which vibrations are applied to a single tendon or muscle using hand-held or wearable devices. Nevertheless, a large number of items will apply to this situation as well. Therefore, as long as no specific reporting guidelines for local vibration therapy exist, and as far as applicable, researchers into local vibration therapy are encouraged to use the reporting guidelines as well.

## 7. Conclusions

The guidelines presented in the current paper should aid researchers, peer-reviewers, and editors to improve the quality and stimulate consistency of reporting about WBV studies. The guidelines extend and progress prior guidelines considerably. More aspects of WBV research are covered, explanations and examples are given, and items specific for preclinical WBV research are included, thus providing new insights into how to report WBV research. The guidelines are suitable for both (clinical) human studies and pre-clinical animal and cell culture studies and can be used next to general reporting guidelines. The guidelines do not include recommendations regarding the application of WBV and should be used in a non-rigid way, with an eye for the large variability in WBV studies.

## Figures and Tables

**Figure 1 biology-10-00965-f001:**
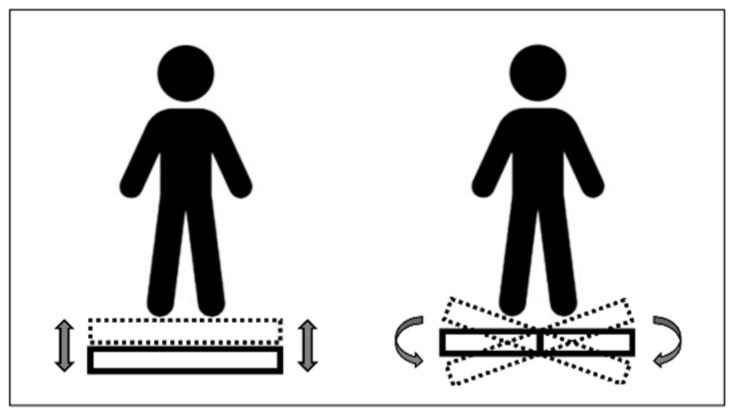
Standing on a platform with vertical vibrations (**left**) versus side-alternating vibrations (**right**).

**Figure 2 biology-10-00965-f002:**
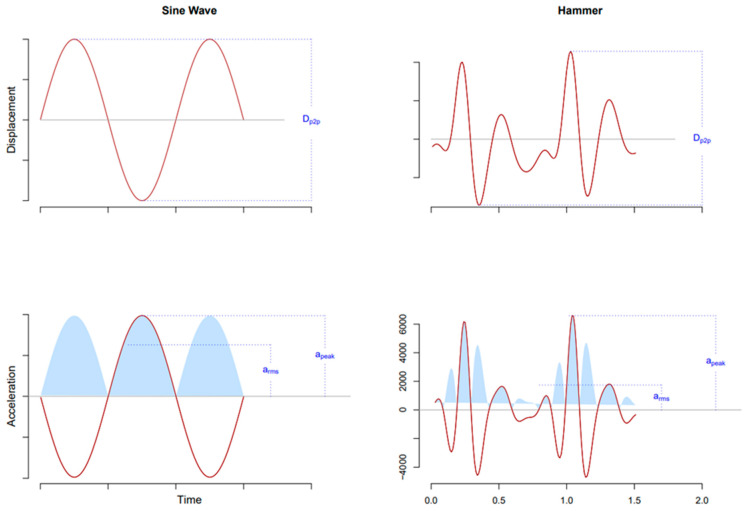
Displacement and accelerations for sine wave and hammer oscillations: displacement and accelerations are only mathematically related for sinusoidal oscillations. Abbreviations: p2p = peak-to-peak; RMS = root-mean-square.

**Figure 3 biology-10-00965-f003:**
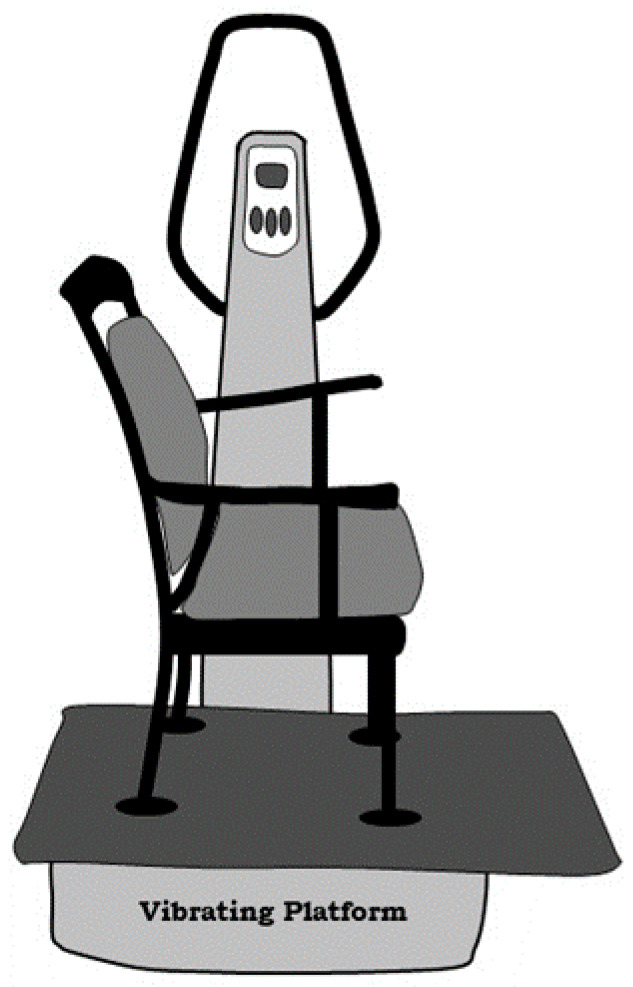
The WBV chair: subjects were sitting on the chair with their feet on the wooden plate and their hands on the arm rests (adopted from Regterschot et al. [11]).

**Figure 4 biology-10-00965-f004:**
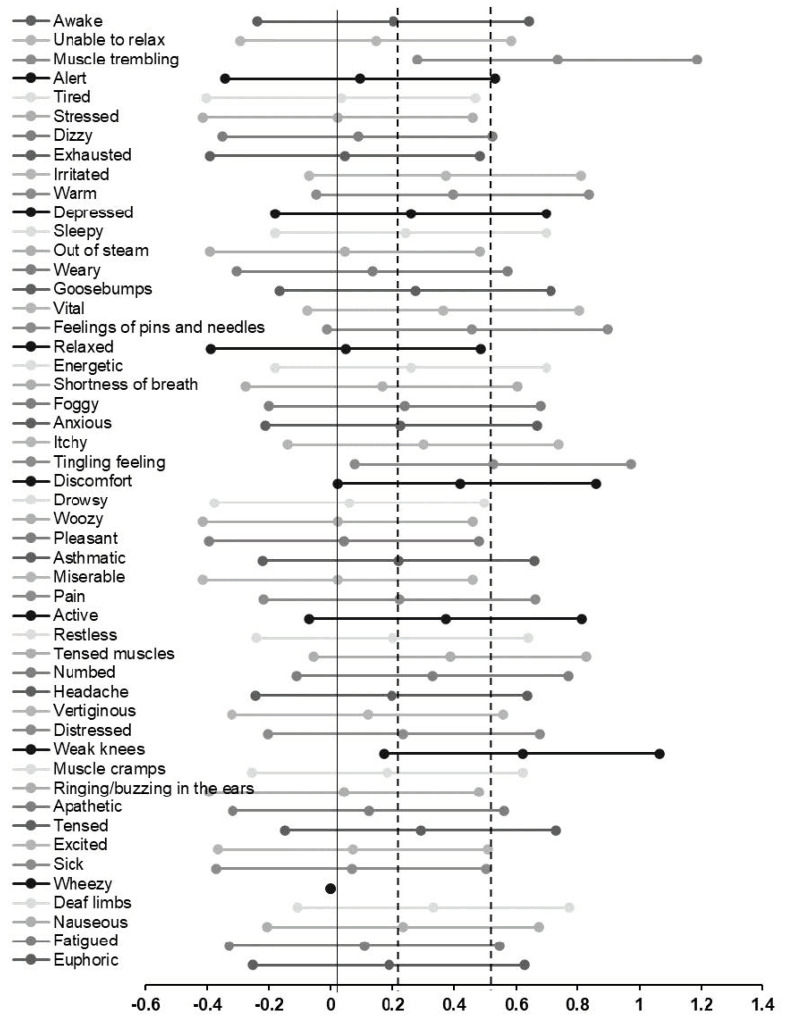
Forest plot of how 88 subjects felt immediately after WBV (intensity of experiences) in effect sizes (Cohen’s d) and 95% CI’s (adopted from Oerlemans et al. [75]).

**Table 1 biology-10-00965-t001:** Checklist of information to include in WBV studies in humans.

**Information about the device**	
*Item*	*Short description*
1	For commercial devices: the manufacturer and production type For commercial and non-commercial (self-built) devices: device specifications
2	If applicable: platform constructions (e.g., mounting a chair on it)
**Information about the vibration**	
*Item*	*Short description*
3	The type of vibration: spatial and temporal characteristics (e.g., vertical or side-alternating)
4	The vibration parameters: definitions and parameter settings used
5	Whether and how the vibration parameters were verified
6	For side-alternating vibrations: the location on the vibration platform where the accelerometer was placed to measure magnitude
7	Whether frequency and magnitude were constant or modulated
**Information about the administration**	
*Item*	*Short description*
8	The posture or body position of the subject and whether it was changed during the intervention (static versus dynamic exercise)
9	The position of the feet of the subjects on the platform during the vibration and how the feet were loaded (e.g., on midfoot or on forefoot with heel lifted off)
10	If and how skidding of feet was prevented
11	If and how vibration transmission to the head was prevented
12	If a handrail was available/used
13	The position of the hands during the WBV and if the hands were directly subjected to the vibration
14	If applicable, the parts of the subjects’ body which were most subjected to vibration (e.g., predominantly the feet)
15	Whether and, if applicable, what tools/aids were used during the vibration (e.g., type and size of dumbbells or resistance bands)
16	General exercise parameters (e.g., duration, number of bouts, rest intervals)
**Information in general protocol**	
*Item*	*Short description*
17	The setting of the WBV sessions/intervention (e.g., hospital, gym, or at home) and the time of day the sessions took place
18	Presence of a trainer (face to face/online) during the WBV sessions
19	The instructions given to the subject before the WBV session
20	Preparatory exercises or warm-up prior to the vibration (type and duration of exercises)
21	The subjects’ footwear (shoes, socks, barefoot) during the vibration, with a detailed description
22	If applicable: characteristics control/sham condition or intervention
23	The moment at which the outcome measures were assessed: during, before, and/or after the vibration; the time between begin/end of exposure/exercise/session and assessment(s)
**Information about the subjects**	
*Item*	*Short description*
24	General characteristics of the subjects
25	The subjects’ previous experience with WBV
26	Acute, short term, or long-term side effects of the vibration exercise

Abbreviation: WBV = whole-body vibration.

**Table 2 biology-10-00965-t002:** Checklist of information to include in WBV studies in animal and/or cell cultures.

**Information about the device**	
*Item*	*Short description*
1	For commercial devices: the manufacturer and production type For commercial and non-commercial (self-built) devices: device specifications
2	The dimensions of the cage or container in which subject(s) was (were) placed during WBV
**Information about the vibration**	
*Item*	*Short description*
3	The type of vibration: spatial and temporal characteristics (e.g., vertical or side-alternating)
4	The vibration parameters: definitions and parameter settings used
5	Whether and how the vibration parameters were verified
6	For side-alternating vibrations: the location on the vibration platform where the accelerometer was placed to measure magnitude
7	Whether frequency and magnitude were constant or modulated
**Information about the administration**	
*Item*	*Short description*
8	Whether the animals were housed individually or in groups during the WBV session ^a^
9	Whether the animals were habituated to the device before the start of the WBV protocol ^a^
10 11	Whether the animals were constrained or not during WBV ^a^ Whether the animals were anesthetized or not during the WBV session ^a^
12	Which posture or body position the animals took on during the vibration (e.g., sitting, standing, lying, depending on the species used) ^a^
13	Whether the position/posture/motor behavior of the animals changed during WBV ^a^
14	Whether other sensory systems were stimulated during the vibration ^a^
15	General exercise parameters (e.g., duration, number of bouts, rest intervals)
**Information in general protocol**	
*Item*	*Short description*
16	The location of the intervention (e.g., a separated experimental room or the room where the animals were housed)
17	The conditions of the test room (e.g., light and temperature)
18	The time of day the WBV session took place (in relation to the light/dark cycle)
19	If applicable: characteristics control/sham/pseudo condition or intervention
20	The moment at which the outcome measures were assessed: during, before and/or after the vibration; the time between begin/end of exposure/exercise/session and assessment(s)
**Information about the subjects**	
*Item*	*Short description*
21	General characteristics of the animals/cell cultures
22	Whether whiskers were present/intact (if species used have whiskers) ^a^
23	The animals’ body weight before and after the entire WBV protocol ^a^
24	Acute, short term, or long-term side effects of the vibrations

Abbreviation: WBV = whole-body vibration. Note: ^a^ only applicable for animal studies.

**Table 3 biology-10-00965-t003:** General exercise parameters that are possibly relevant for WBV exercise.

WBV Program/Intervention	WBV Session ^a^
Program duration (weeks)	Mode (with/without additional exercise)
Frequency (sessions week^−1^ and/or sessions day^−1^)	Session duration (min) ^b^
Attendance (%)	Number of WBV bouts
	Duration of WBV bouts (min)
	Rest between WBV bouts (min)
	Intensity additional exercise (absolute (kg) or relative (% 1RM or % 5RM) or, for aerobic additional exercise, HR or HR as % of maximal HR)
	Number of sets/repetitions of additional exercise

^a^ For frequency and magnitude vibrations, see Items 7–8. ^b^ May include warm-up and measurements next to one or more bouts of WBV. Abbreviations: WBV = whole-body vibration; HR = heart rate.

## Data Availability

Not applicable.

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
