# Peer review of "Reporting Guidelines for Whole-Body Vibration Studies in Humans, Animals and Cell Cultures: A Consensus Statement from an International Group of Experts"

_biology, 2021, doi:10.3390/biology10100965_

Round 1
Reviewer 1 Report
First, I would like to thank you for the opportunity to review this nice manuscript. The design is strange because I have never seen a tutorial as a research paper design.
However, even though I think the topic is widely studied and I have my doubts as a researcher about the potential of vibration at the clinical or cellular level, the manuscript is good and perfectly done. I would suggest adding more pictures to make it more comfortable to read as well as summarising some paragraphs of the items because it gets really long.
Thank you for the contribution to the state of the art.
Reviewer 2 Report
Thank you for your submission.
The manuscript reports guidelines related to whole-body vibration. A tutorial was conducted to update guidelines related to whole-body vibration studies in humans, animals and cell cultures. This manuscript is clear and concise, and the authors show a revealing command of updated information that fellow authors should be including in published manuscripts. The authors have done a great job of providing an informative and meaningful addition to the current study field. There are a small few minor comments that may need clarification.
Introduction:
Have all WBV studies reported improvements then? Or are there studies whereby there was no positive effect found...
line 68 - dynamic exercises - might be beneficial to add an example for the reader here
Methods:
Point 4. These results have already been registered in EQUATOR then?
Checklists look good to me
For device specifications - is there a low/high filter applied to any of the data extracted or within the device itself that may need to be reported?
line 245 - space
Item 16 - is WBV considered an aerobic exercise in nature then?
Would heart rate or metabolic rate or such ever be measured?
Item 17 and the describing sentence should be in bold?
What about circadian rhythm having an influence on WBV?
Very nicely written discussion and conclusion.
Reviewer 3 Report
This article adds considerably to the research involving WBV. This article provides researchers direct information that will improve the quality and stimulate consistency of reporting WBV studies.
Very valuable to have this sort of article for anyone doing WBV research.
